# Changes in Phenols, Polysaccharides and Volatile Profiles of Noni (*Morinda citrifolia* L.) Juice during Fermentation

**DOI:** 10.3390/molecules26092604

**Published:** 2021-04-29

**Authors:** Zhulin Wang, Rong Dou, Ruili Yang, Kun Cai, Congfa Li, Wu Li

**Affiliations:** 1Key Laboratory of Food Nutrition and Functional Food of Hainan Province, College of Food Science and Engineering, Hainan University, Haikou 570228, China; 18085231210024@hainanu.edu.cn (Z.W.); dourong@yili.com (R.D.); caikun@hainanu.edu.cn (K.C.); congfa@hainanu.edu.cn (C.L.); 2College of Food Science, South China Agricultural University, Guangzhou 510642, China; rlyang77@scau.edu.cn

**Keywords:** noni juice, fermentation, volatile compound, phenols, polysaccharides, GC-MS, E-nose

## Abstract

The change in phenols, polysaccharides and volatile profiles of noni juice from laboratory- and factory-scale fermentation was analyzed during a 63-day fermentation process. The phenol and polysaccharide contents and aroma characteristics clearly changed according to fermentation scale and time conditions. The flavonoid content in noni juice gradually increased with fermentation. Seventy-three volatile compounds were identified by solid-phase microextraction coupled with gas chromatography–mass spectrometry (SPME-GC-MS). Methyl hexanoate, 3-methyl-3-buten-1-ol, octanoic acid, hexanoic acid and 2-heptanone were found to be the main aroma components of fresh and fermented noni juice. A decrease in octanoic acid and hexanoic acid contents resulted in the less pungent aroma in noni juice from factory-scale fermentation. The results of principal component analysis of the electronic nose suggested that the difference in nitrogen oxide, alkanes, alcohols, and aromatic and sulfur compounds, contributed to the discrimination of noni juice from different fermentation times and scales.

## 1. Introduction

Noni (*Morinda citrifolia* L.), is a plant that grows widely in tropical and subtropical regions around the world [1,2]. The roots, stems, leaves, fruits and seeds of noni have been used as traditional medicine for over a thousand years to treat diabetes, headaches and arthritis [3,4]. As the edible part of the plant, noni fruits contain polysaccharides, polyphenol and other bioactive ingredients [5,6,7,8]. Pharmacological research indicated that noni fruits exhibited therapeutic effects on inflammation, diarrhea, hypertension and hypoglycemia [9,10,11].

Although previous reports have shown that noni fruit possesses a variety of biological activities [1], the bad taste and smell of noni fruit make it unsuitable for fresh consumption. After fermentation, noni juice becomes partly free from the undesirable smell. In addition, the antioxidant activity of noni juice was found to be significantly increased after fermentation [12]. Recent studies showed that fermented noni juice can improve acute alcohol-induced liver injury [13] and regulate immunity response [14].

In previous reports, approximately 80 volatile compounds, including fatty acid esters, monoterpenes, acids, alcohols and aldehydes, were detected in fresh noni fruit. The dominant volatile compounds were found to be octanoic acid and hexanoic acid [8,15]. Octanoic acid, hexanoic acid, and butanoic acid, which are considered to be linked to cheese, strong sweat, and faint and irritating odors [16], may be the primary contributors to noni juice’s unpleasant odor [17].

At present, noni juices produced over different fermentation periods are available on the market, which can be grouped into short-term, month-long, and long-term fermentations [18]. However, the flavor profiles of fermented noni juice with different fermentation times are not clear. In addition, as a natural fermentation, the scale of fermentation may have an impact on the active substances and flavor of fermented juice [19]. The characterization of the noni flavor profile may be different between laboratory- and industrial-scale fermentation.

The aim of this study was to explore the change in volatile compounds, phenols and polysaccharides of noni juice produced under laboratory- and industrial-scale fermentation conditions. These results may provide a basis for improving fermented noni juice production.

## 2. Results and Discussion

### 2.1. Changes in Polyphenols, Flavonoids and Polysaccharides during Fermentation

The total phenolic content of noni juice produced over different fermentation times is shown in Figure 1a. The total phenolic content of the fresh juice (0 day, F0) was 168.35 ± 3.54 mg GAE/100 mL. Over the 63-day fermentation, the total phenolic content first decreased, then increased, before decreasing again. The concentration of total phenols reached its lowest level in L56 and F21 (125.93 ± 12.02 mg GAE/100 mL and 156.02 ± 6.20 mg GAE/100 mL, respectively). The highest total phenolic contents were 199.77 ± 8.12 mg GAE/100 mL and 185.60 ± 5.20 mg GAE/100 mL in L28 and F28, respectively. The contents of polyphenols showed significant changes in both the laboratory- and factory-scale fermentation process. Moreover, there were significant differences in the phenolic contents on day 7, 14, 21, 35, 49, 56 and 63 of fermentation between the laboratory- and factory-scale fermentation.

As shown in Figure 1b, the flavonoid content showed an increasing trend during the 63-day fermentation of laboratory- and factory-scale fermentation processes. Compared with the fermented juice, the flavonoid content of the fresh noni juice was the lowest (26.07 ± 0.54 mg RE/100 mL). The flavonoid content reached its highest value of 36.73 ± 2.51 mg RE/100 mL on day 63 of laboratory-scale fermentation. The highest flavonoid content of factory-scale fermentation was 43.28 ± 1.66 mg RE/100 mL on day 49, after which it decreased. During the whole fermentation process, the flavonoid content of factory-scale fermentation was commonly higher than the laboratory-scale fermentation. After 35 days of fermentation, there was a significant difference in flavonoid content between the two fermentation scales.

During the fermentation of juice, the species and quantity of microorganisms contribute to the sensory quality and the transformation of bioactive components [20,21]. Microorganisms metabolize food and transform polyphenols through complex enzyme systems during juice fermentation [22]. Previous studies showed that the contents of hydroxycinnamic acids and anthocyanins were decreased and the total concentration of flavanols was increased during black currant juice fermentation [23]. The total content of flavonoids was reported to increase during *Dendrobium officinale* and banana juice fermentation by lactic acid bacteria [24]. The change of polyphenols and flavonoids content may be due to the intricate polyphenols being broken down by enzymes into simpler flavanols during fermentation [25,26]. This result is consistent with the observations of Kelanne et al. [23], who reported an increase in flavanols and a decrease in total content of polyphenols during yeast fermentation.

As shown in Figure 1c, the content of polysaccharides in fresh noni juice (day 0) was 6.45 ± 0.71 mg/mL. The content of polysaccharide showed a downward trend during the first 42 days of fermentation and reached a low of 2.11 ± 0.71 mg/mL on day 42, then increased to a high of 6.28 ± 2.48 mg/mL on day 56 of the laboratory-scale fermentation. During the factory-scale fermentation, the polysaccharide content initially increased but then decreased. The polysaccharide content reached the highest value of 8.66 ± 0.30 mg/mL on day 7 and decreased to relatively low values, at day 42 to 63, of 4.61 ± 0.96 mg/mL–3.57 ± 1.10 mg/mL.

Polysaccharide is considered to be one of the main bioactive components of fermented noni juice [1]. Hydrolytic enzymes produced by microorganisms can degrade the insoluble pecto-cellulosic cell wall into soluble polysaccharides and break polysaccharides down into oligosaccharides or monosaccharides, resulting in an increase or decrease in polysaccharide concentration during fermentation [27,28]. The content of polysaccharide was found to be increased in the fermentation of carrot juice by *Lactobacillus gasseri*, which synthesized fructosyltransferase enzymes during fermentation that enabled the conversion of simple sugars, primarily into polysaccharides [29]. Meanwhile, *Lactobacillus plantarum* reduced the content of polysaccharide during the fermentation process of five fruit juices [30]. The effects of different microorganisms to release cell wall polysaccharides showed significant differences under the same fermentation conditions [31].

The dominant microorganisms were influenced by fermentation conditions, which could result in changes in the enzyme system and matrix metabolism. Bergh et al. [19] reported that there were significant differences in the aroma, color and turbidity between laboratory- and factory-scale fermentation of *Cyclopia intermedia*. In this study, the different fermentation scales may have caused the change of fermentation density and temperature, which resulted in differences between the two fermentation scales. Furthermore, our results (unpublished) showed that the microbial profile of fermented noni juice differed significantly between the laboratory- and factory-scale. In short, our results show that the contents of phenolic, flavonoid and polysaccharide showed significant differences between the laboratory- and factory-scale fermentation processes, which may be due to the different microbial compositions caused by different fermentation scales.

### 2.2. Characterization of Volatile Compounds

As shown in Figure 2 and Figure 3, significant differences in volatile components were present in fresh noni juice (F0) and fermented noni juice produced under different fermentation time and scale conditions. A total of 73 volatile compounds were detected in fresh and fermented noni juice (Figure 3). The volatile compounds were divided into 8 species: esters, alcohols, acids, ketones, aldehydes, alkanes, amines and phenols. There were 48, 57, 49, 45 and 37 volatile compounds detected from the F0, L7, L28, L63 and F63, respectively. Additionally, esters, alcohols, acids and ketones were found to be the major constituents of total volatile compounds, accounting for more than 98% of all samples. Among those, acids were the predominant component, accounting for 76.45, 53.39, 81.45, 84.71 and 49.64% of the total volatile compounds in F0, L7, L28, L63 and F63, respectively.

The details of the 73 compounds identified are listed in Table 1. Twenty-five compounds were detected in fresh and fermented samples, which included twelve esters (ethyl acetate, methyl butyrate, ethyl butyrate, 3-butene-3-methyl-methyl ester, methyl hexanoate, ethyl hexanoate, 4-pentenyl butyrate, methyl octanoate, 3-methyl-3-butenoate, ethyl-2-(5-methyl-5-vinyl tetrahydrofuran-2-yl)-2-propyl-carbonate, 3-(methylthio) methyl propionate and methyl salicylate), six alcohols (3-methyl-3-buten-1-ol, 3-methyl-2-buten-1-ol, 2-heptanol, 1-hexanol, 1-octene-3-ol, and 3,7-dimethyl-1,6-octanethiol-3-ol), four acids (butyric acid, 2-methylbutyric acid, hexanoic acid and octanoic acid) and three ketones (2-pentanone, 2-heptanone and 2-nonanone). Compared with fresh noni fruit, there were eighteen new volatile compounds, including methyl acetate, ethyl 2-methylbutyrate, methyl phenyl acetate, acetic acid and 2-butanone, at day 7 of fermentation in the laboratory. Four new volatile compounds (2-methyl-3-butene-2-ol, 2-pentenoic acid, benzaldehyde and eugenol) were detected in the sample of L28. These 25 compounds contributed to the common volatile compounds of fresh and fermented noni juice. Compared with L28, the new compounds formed in L63 were methyl 2-methyl butyrate, 2,6,11-trimethyl-dodecane, 2,4-dimethylbenzaldehyde, erucamide and oleic acid amide. Previous research showed that benzaldehyde [32] and ethyl 2-methylbutyrate [33] play certain roles in the formation of the flavor of fermented juice. The data obtained showed that the contents and varieties of sulfur-containing esters (such as amyl 3-(methylthio)propionate) decreased over the course of the fermentation progress. These sulfur compounds may have a negative effect on the organoleptic characteristics of juice, which generally produce unpleasant odors and tastes described as resembling garlic, onion, cabbage or sulfur [34].

Acids were the largest volatile component in fresh and fermented noni juice. Octanoic acid and hexanoic acid were found to be the most significant acid compounds in all the samples, accounting for 49.32–67.30% and 24.58–44.58% of the total acids, respectively. At day 7 of the fermentation, the octanoic acid concentration decreased to 28.88% of the total volatile compounds and then increased to the level of fresh fruit over time; hexanoic acid showed a similar trend. In addition, the hexanoic acid content at F63 (12.20%) was significantly lower than L63 (36.93%). Despite the relatively high odor threshold values of octanoic acid and hexanoic acid (910 and 420 μg/L [16,35], respectively), the comparatively high concentrations of them in fermented noni juice resulted in a high odor activity value (OVA) in our study. The enrichment of octanoic acid and hexanoic acid contributed to the characteristic odor of fermented noni juice, with odors described as similar to cheese and sweat, with faint and irritating characteristics [16,17,36]. The decrease in octanoic acid and hexanoic acid content in F63 may have contributed to its less pungent aroma in noni juice produced from factory-scale fermentation.

Esters play an important role in the flavor of juice [37,38]. In this study, the methyl hexanoate (with an odor threshold of 87 μg/L [39]) was found to be the quantitively predominant ester compound in fermented noni juice. Although the ester content did not show a wide increase in fermented noni juices (Figure 3b), the content of methyl hexanoate and methyl salicylate, which were characterized by positive floral and fruity aromas [37,40], increased during fermentation. The relative content of methyl hexanoate in L7, L28 and F63 increased the content of F0 by 28.93, 4.06 and 69.04% in L7, L28 and F63, respectively. In addition, the content of methyl hexanoate in F63 was higher by 139.57% than that in L63, which could contribute to the improvement of the flavor of factory-fermented noni juice.

As shown in Table 1, the main alcohols were 3-methyl-3-buten-1-ol, 1-hexanol, ethanol and 2-heptanol. Under laboratory-scale fermentation, the content of 3-methyl-3-buten-1-ol, 1-hexanol and 2-heptanol increased on day 7 of fermentation and then decreased. The content of 3-methyl-3-buten-1-ol (odor threshold 550 μg/L), which gave the juice an herbaceous aroma [41], increased from 3.52 % (F0) to 5.57% (L7). The content of 1-hexanol, described as a “sweet alcohol” aroma [16], increased sharply from 0.05% (F0) to 18.31% (L7) in laboratory-scale fermentation, and dropped to a low of 0.04% at 63 days. The 2-heptanol (odour threshold 65 μg/L) increased by fermentation and endowed juices with the odor of citrus, mushroom and herbs [42]. These results suggest that the flavor of noni juice may be improved in the early stage of laboratory-scale fermentation. In addition, the content of the main alcohols was different in factory-scale (F63) and laboratory-scale (L63) fermentation.

Seven kentones were detected in fresh and fermented noni juice. Their content was relatively low, except for 2-heptanone (odor threshold 140 μg/L [39]), which contributed “milk powder”, “potato” and “cheese-like” odors [16]. The content of 2-heptanone presented a trend of increasing first then decreasing, with content of 3.64, 6.59, 6.71, and 3.59%, at F0, L7, L28, L63, respectively. The content of 2-heptanone in factory-scale fermentation was 39.82% which was significantly different to that of laboratory-scale fermentation.

Different microbial communities give different flavors during the fermentation process [43]. The microflora responsible for fermentation contributes to the aroma by utilizing constituent ingredients of plants to produce enzymes that transform natural compounds into flavor compounds [44]. Lactic acid bacteria turned the carbohydrates of fermented minced pepper into organic acids and alcohols during the fermentation process, while the organic acids and alcohols could combine with each other to form several esters, such as ethyl acetate, with pineapple flavor [45]. Lee et al. [46] reported that benzaldehyde, which has a cherry-like aroma, was produced from phenylalanine by *Tetagenoccocus halophilus* degradation in soy sauce. In this study, the volatile profile obviously differed between laboratory- and factory-scale fermentation. The relative contents of acids, alcohols, phenols and ketones of L63 were 63.08, 220.94, 170.65 and 8.89% of F63, respectively. Meanwhile, several volatile compounds identified in L63 were not detected in F63, and vice versa. The 2-heptanone was the dominant aroma compound in F63, reaching a relative concentration above 39.82%, and this may be responsible for the strongly pleasant aroma, while its content was 3.59% in L63. In short, the results of this study showed that the flavor improved in the early stages of laboratory-scale fermentation and became pungent in the late stages of fermentation. At the same time, the result of volatile profile analysis suggested that the flavor of noni juice of factory-scale fermentation may be preferred to that of laboratory-scale fermentation.

### 2.3. Electronic Nose Analysis

Electronic nose (E-nose) analysis was performed to evaluate the differences in volatile profiles between samples. A typical E-nose radar plot of the samples is presented in Figure 4. The response value of the S1, S3 and S5 sensors, which are selective toward aromatic compounds (Table 2), increased in the samples of laboratory- and factory-scale fermentation. The results suggested that fermentation processing could result in the formation of aromatic compounds. The decrease in response values of the S2, S6, S7 and S9 sensors indicated that nitrogen oxide, alkanes and sulfides decreased during the fermentation process. Compared with F0, the response values of F63 and L63 exhibited similar trends. Furthermore, the response values of S6, S8 and S9 of F63 were significantly higher than those of L63, which suggested that more alkanes, alcohols and aromatic compounds were produced during factory-scale fermentation. The response values of S7 and S9 sensors were significantly reduced in the laboratory- and factory-scale fermentation processes. These results suggest that fermentation processing could lead to a reduction in sulfides, which is consistent with the results of GC-MS.

The principal component analysis (PCA) for E-nose analysis is shown in Figure 5. From Figure 5a, it can be observed that two principal components (PC1 81.15% and PC2 11.00%) accounted for 92.15% of the total variance. The fresh noni juice and the samples with different fermentation times and scales were divided into four quadrants, which indicated that there was an obvious difference in the flavor profiles of the samples. In general, these results suggested that the E-nose can characterize the samples produced under different fermentation time and scale conditions. Figure 5b shows the score plot of the 10 E-nose sensors, which indicates the degree of contribution each E-nose sensor made to sample discrimination. PC1 had high contributing factor loadings from S2, S4, S6, S7, S8 and S9 sensors, which helped to discriminate samples F0 from L28, L63 and F63. PC2 had high contributing factor loadings from the S10 sensor. The results of GC-MS analysis showed that F0 had a higher concentration of sulfur compounds (methyl 3-(methylthio)propionate, ethyl 3-(methylthio)propionate and amyl 3-(methylthio)propionate) than L28, L63 and F63, which were mainly associated with the response of sensors S7 and S9. F63 had a higher content of 2-heptanone than F0, L28 and L63, which was associated with sensors S8 and S9. L63 had a higher content of 2,6,11-trimethyl-dodecane, indicating that the S6 and S10 may be sensitive to 2,6,11-trimethyl-dodecane in L63. Combined with the results of GC-MS, the results of the PCA showed that the nitrogen oxide, alkanes, alcohols, and aromatic and sulfur compounds, contributed to distinguishing the volatile profiles of different samples. Although the E-nose does not allow for identification of individual volatile compounds, the sensor signals present variable responses for the samples and give volatile fingerprints, which facilitate good discrimination of noni juice produced under different conditions [47,48]. Moreover, the E-nose analysis data contain synthesized information rather than simple qualitative and quantitative results of individual volatile compounds, and the analysis is faster and less expensive than gas GC-MS [48,49].

## 3. Materials and Methods

### 3.1. Materials and Reagents

Noni fruits (cultivars “Kuke”) were provided by the Hainan Wanning Noni Industrial Base (Hainan, China) in March 2017. Gallic acid (purity, 98%) and rutin (purity, 97%) were purchased from Alfa Aesar (Ward Hill, MA, USA). Phenol and Folin–Ciocalteu were purchased from Solarbio Technology Co., Ltd. (Beijing, China). The 3, 5- dinitrosalicylic acid (purity, 98%) was purchased from Sinopharm Chemical Reagent Co. Ltd. (Shanghai, China). All other chemical reagents were analytically pure.

### 3.2. Noni Fruit Fermentation

#### 3.2.1. Factory-Scale Fermentation

The fermentation was performed according to the factory procedure. The same maturity noni fruits were selected and cleaned with running water. Noni fruits were soaked with 60 mg/L chlorine dioxide for 30 min and then rinsed with sterile water. After air drying at room temperature, the noni fruits were canned for fermentation in a 10-ton fermenter at room temperature. The fermented juice was collected from the factory fermentation tank on days 7, 14, 21, 28, 35, 42, 49, 56 and 63 (F7-F63) and stored at −80 °C.

#### 3.2.2. Laboratory-Scale Fermentation

The noni fruits in the same batch as those used for factory fermentation were fermented in a laboratory according to the factory procedure. Briefly, the noni fruits were cleaned with running water and then soaked with 60 mg/L chlorine dioxide for 30 min. After being cleaned with sterile water, the fruits were put in a 6 L sterile fermenter for fermentation at room temperature. The fermented juice was collected on days 7, 14, 21, 28, 35, 42, 49, 56 and 63 (L7−L63) and stored at −80 °C.

### 3.3. Determination of Polyphenols and Flavonoids

The total phenols content was determined using the Folin–Ciocalteu method as described by Wolfe et al. [50]. Briefly, a volume of 0.625 mL of juice was diluted 50-fold with pure water and added to a test tube. Then, 0.125 mL of Folin–Ciocalteu reagent was added to the solution. After 6 min, the solution was mixed with 1.25 mL of 7.0% Na_2_CO_3_ and 1 mL pure water and allowed to react in the dark for 90 min. Finally, the absorbance of the solution was measured at 760 nm using a ST-360 ELISA spectrophotometer (Shanghai Kehua Bio-Engineering co. LTD, Shanghai, China). The total phenols content was quantified using the gallic acid standard curve (0–250 mg/100 mL) and was expressed as milligrams of gallic acid equivalents (GAE) per 100 mL of noni juice.

The total flavonoids content was determined using the method described by Dewanto et al. [51]. Briefly, a sample of 0.3 mL was mixed with 1.5 mL pure water and 90 μL 5.0% NaNO_2_ solution. After 6 min, 180 μL of 10.0% AlCl_3_ was added and allowed to stand for 5 min. Then, 0.6 mL of 1 M NaOH was added and the solution was brought to 3 mL with pure water. The absorbance was measured at 510 nm using a microplate reader (SpectraMax 190, Molecular Devices, Sunnyvale, CA, USA) and compared to the standards prepared equally with known rutin concentrations (0–100 mg/100 mL). The results were expressed as milligrams of rutin equivalents (RE) per 100 mL of noni juice.

### 3.4. Determination of Polysaccharides

The phenol-sulfuric acid method [52] and dinitrosalicylic acid (DNS) colorimetric method [53] were used to analyze the total sugar content and the reducing sugar content, respectively. The polysaccharide content was equal to the total sugar content minus the reducing sugar content.

### 3.5. GC-MS Analysis

SPME-GC-MS analysis was performed according to the method described by Pino et al. [17], with appropriate modifications. The parameters for volatile compound analysis were as follows: The samples were swirled at 1400 rpm for 3 min; the 5 mL sample and 4 mL saturated NaCl solution were blended in a 250 mL head space bottle sealed with a parafilm, shaken well and bathed at 40 °C for 30 min. The parafilm was pierced with the sample injector of SPME equipped with a 2 cm coated fiber of carboxen/divinylbenzene/polydimethylsiloxane (DVB/CAR/PDMS) (coating thickness: 50/30 μm), and the SPME fiber was exposed to the sample headspace for 10 min. Afterwards, the fiber was plugged into the injection port of the GC and rapidly desorbed at 240 °C for 3 min.

The GC-MS analyses were carried out by a Shimadzu GC-MS45-QP2010 Ultra Series GC-MS system (Shimadzu, Kyoto, Japan) equipped with a DB-Wax (30 m × 0.25 mm × 0.25 μm) elastic silica capillary column. The GC temperature program was maintained at 40 °C for 10 min, and followed by a temperature increment of 8 °C/min to 240 °C. High purity nitrogen was used as carrier gas, with a flow rate of 1 mL/min. The mass spectrometer was used in electron impact mode with an electron energy of 70 eV. The data obtained were searched by the NIST library and the relative content of each chemical component was measured by the peak area normalization method.

### 3.6. Electronic Nose Analysis

The electronic nose (E-nose) equipment (PEN 3 Portable E-nose, Airsense Analytics GmbH, Schwerin, Germany) consisted of 10 diverse metal oxide semiconductor sensors. The main properties of the 10 metal oxide sensors are shown in Table 2. In response to the analysis of E-nose, each sample (10 mL) was placed in a 50 mL glass bottle and covered with a parafilm. The sample was incubated for 60 min at room temperature to allow for the evolution of headspace. In subsequent tests for the volatile compounds of headspace, the time of auto-zeroing and detection was 10 s and 180 s, respectively. The headspace gas was pumped into the sensor chamber at a flow rate of 400 mL/min, and the recovery time for the sensors was 170 s.

### 3.7. Statistical Data Analysis

Experimental data are expressed as means ± standard deviations and were analyzed using SPSS version 22 software (SPSS Inc., Chicago, IL, USA). Analysis of variance (ANOVA) tests were used, and statistical significance was set at *p* < 0.05.

## 4. Conclusions

The present study indicated that there were significant differences in the contents of phenols and polysaccharides and volatile profiles of noni juice produced under different fermentation time and scale conditions. The flavonoid content in noni juice showed an increasing trend throughout the fermentation process and reached its highest level at L63 and F49 (36.73 ± 2.51 mg RE/100 mL and 43.28 ± 1.66 mg RE/100 mL, respectively). A total of 73 volatile compounds were identified in fresh and fermented noni juice, in which methyl hexanoate (1.39–3.33%), 3-methyl-3-buten-1-ol (0.47–5.57%), octanoic acid (28.88–43.75%), hexanoic acid (12.20–36.93%) and 2-heptanone (3.59–39.82%) were the major aroma components. The contents of hexanoic acid and octanoic acid in F0 and F63 decreased from 34.08 and 39.69% to 12.20 and 33.41% in noni juice after factory fermentation, respectively, which resulted in a less pungent aroma. To the best of our knowledge, this is the first report on the changes of volatile profiles in noni juice during the fermentation process. The results of the present study may contribute to the improvement of noni juice quality.

## Figures and Tables

**Figure 1 molecules-26-02604-f001:**
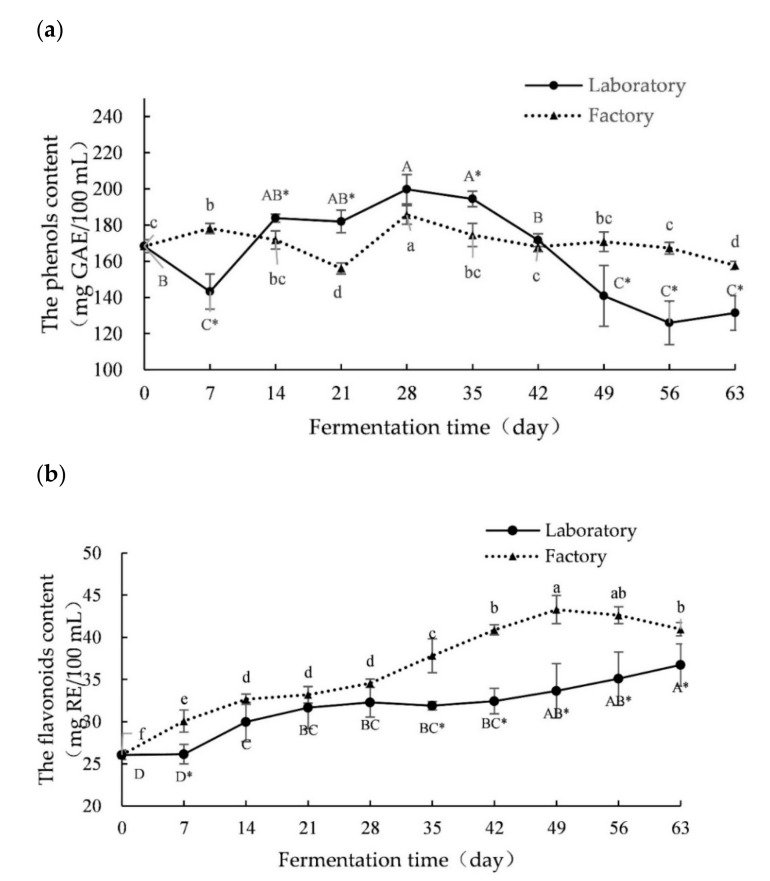
Changes in the total phenols (**a**), flavonoids (**b**) and polysaccharides (**c**) content during laboratory- and factory-scale fermentation. Different letters in the same line represent significant differences at *p* < 0.05. * represents a significant difference between the laboratory and the factory at the same fermentation time point (*p* < 0.05).

**Figure 2 molecules-26-02604-f002:**
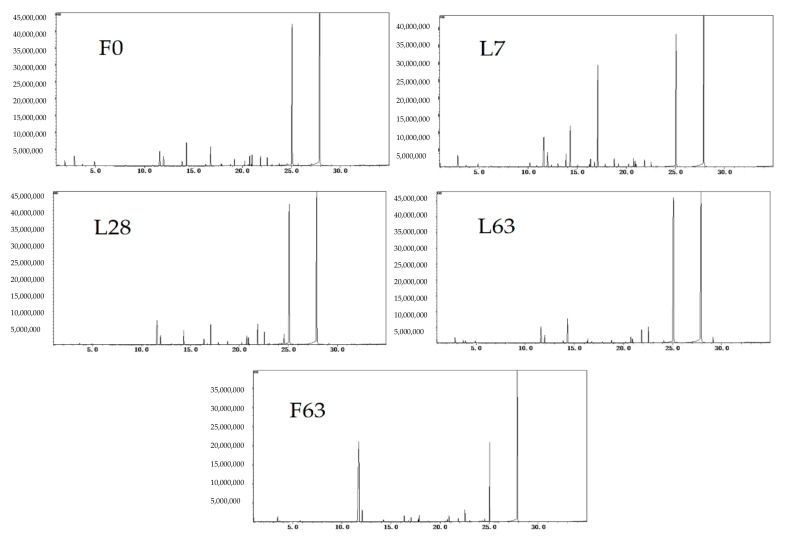
GC-MS total ion chromatogram of the volatile components of noni juice at different fermentation times and scales.

**Figure 3 molecules-26-02604-f003:**
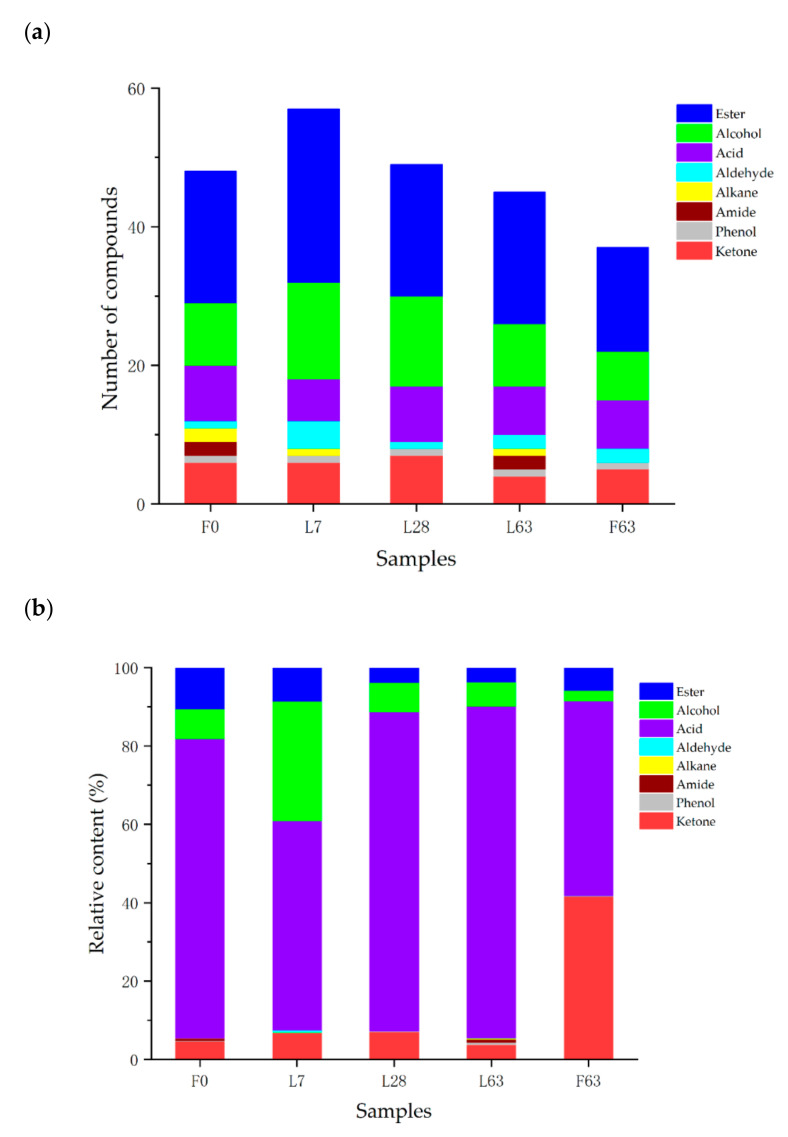
Volatile compounds identified by SPME-GC–MS of noni juice with different fermentation times and scales. (**a**) Number of volatile compounds. (**b**) Relative content of volatile components.

**Figure 4 molecules-26-02604-f004:**
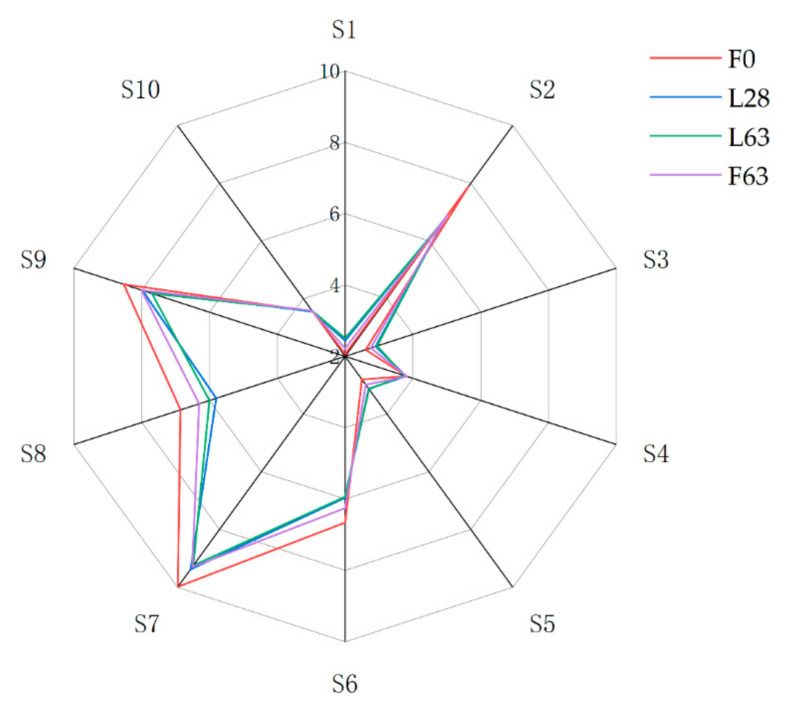
Radar plot of the volatile compounds of noni juice at different fermentation times and scales by E-nose. The data are normalized as a result of LOG_2_ (mean value × 10).

**Figure 5 molecules-26-02604-f005:**
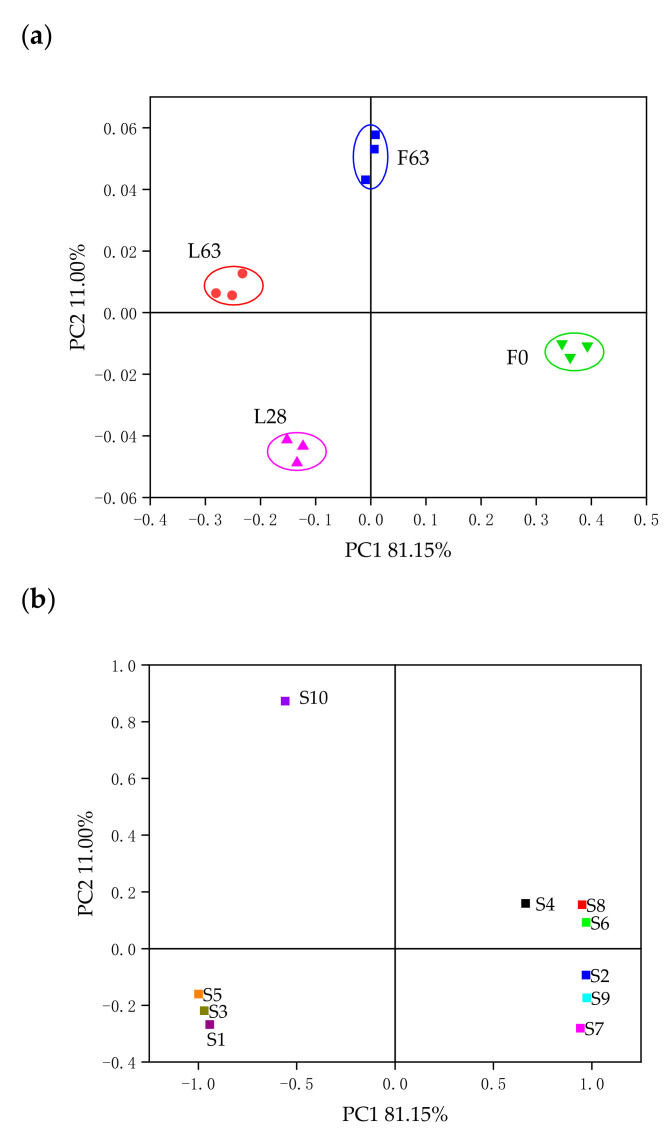
PCA results for E-nose analysis of different samples. Loading plot (**a**); score plot (**b**).

**Table 1 molecules-26-02604-t001:** Volatile compounds of noni juice at different fermentation times and scales.

Compounds	Rention Time/min	Relative Content (%)				
		F0	L7	L28	L63	F63
**Esters (28)**						
methyl acetate	1.93	nd	0.03	0.07	0.03	0.06
ethyl acetate	2.35	0.02	0.08	0.05	0.06	0.12
Methyl butanoate	3.66	0.34	0.29	0.22	0.44	0.21
methyl 2-methylbutyrate	4.18	nd	nd	nd	0.01	0.03
ethyl butyrate	4.88	0.70	0.48	0.12	0.28	0.04
vinyl butyrate	5.07	nd	nd	nd	nd	0.24
ethyl 2-methylbutyrate	5.36	nd	0.02	0.01	0.02	nd
3-butenoic acid-3-methyl-methyl ester	8.07	0.21	0.22	0.05	0.06	0.10
methyl hexanoate	11.93	1.97	2.54	2.05	1.39	3.33
3-methyl-3-buten-1-ol acetate	12.36	0.13	0.33	0.03	0.05	nd
butyl butyrate	13.28	nd	0.10	nd	nd	nd
ethyl hexanoate	13.82	0.77	1.71	0.12	0.36	0.03
3-methyl-2-buten-1-ol acetate	14.42	0.02	0.04	nd	nd	nd
4-pentenyl butyrate	14.94	2.40	0.66	0.20	0.19	0.31
hexyl acetate	15.03	nd	0.28	nd	nd	nd
methyl octanoate	17.84	0.25	0.32	0.27	0.15	1.03
3-methylbutyl-2-alkenyl butyrate	17.9	0.17	0.04	0.01	0.01	nd
ethyl octanoate	18.79	0.20	nd	nd	nd	nd
3-methyl-3-butenoate	19.18	0.77	0.34	0.04	0.02	0.05
ethyl-2-(5-methyl-5-vinyltetrahydrofuran-2-yl)-2-propyl-carbonate	19.33	0.02	0.01	0.01	0.03	0.02
methyl 3-(methylthio)propionate	20.23	0.60	0.33	0.26	0.15	0.11
ethyl 3-(methylthio)propionate	20.98	1.25	0.29	0.09	0.09	nd
3-methylbutyl-3-alkenyl isobutylate	23.75	0.25	0.01	nd	nd	nd
methyl phenylacetate	23.90	nd	0.02	0.02	0.03	nd
methyl salicylate	24.10	0.04	0.09	0.16	0.27	0.09
hexyl octanoate	24.90	nd	0.23	nd	nd	nd
butyric acid, di(tert-butyl)silyl ester	24.98	nd	0.09	nd	nd	nd
amyl 3-(methylthio)propionate	25.7	0.37	0.02	nd	nd	nd
**Alcohols (15)**						
ethanol	2.82	2.44	2.17	0.12	1.20	nd
2-methyl-3-buten-2-ol	5.05	nd	nd	0.12	0.07	nd
2-methyl-1-propanol	7.46	nd	0.10	nd	nd	nd
1-butanol	10.17	nd	1.05	0.06	nd	nd
2-methyl-1-butanol	13.02	nd	0.63	0.20	nd	nd
3-methyl-3-buten-1-ol	14.24	3.52	5.57	2.17	3.30	0.47
3-methyl-2-buten-1-ol	16.24	0.21	0.34	0.12	0.20	0.07
2-heptanol	16.35	0.08	0.82	0.74	0.46	0.93
1-hexanol	17.04	0.05	18.31	2.67	0.04	0.68
1-octene-3-ol	19.04	0.01	0.01	0.04	0.14	0.04
3,7-dimethyl-1,6-octanethiol-3-ol	20.74	1.08	0.73	0.97	0.61	0.38
1-octanol	20.9	nd	0.58	nd	nd	nd
alpha-terpineol	23.03	0.18	0.09	0.22	nd	0.20
3-(methylthio)-1-propanol	23.22	nd	0.10	0.01	nd	nd
benzyl alcohol	25.45	0.02	0.04	0.06	0.10	nd
**Acids (10)**						
cyclopentadecandecanoic acid	1.02	0.04	nd	nd	nd	nd
cyclohexanecarboxylic acid	10.82	0.23	0.35	0.08	0.13	nd
acetic acid	18.76	nd	1.21	0.53	0.35	0.19
sarnic acid	20.10	0.07	nd	nd	nd	nd
2-methylpropionic acid	20.90	0.17	nd	0.89	0.46	0.95
butyric acid	21.85	1.14	0.85	2.57	1.41	0.53
2-methylbutyric acid	22.53	1.03	0.50	1.59	1.68	1.88
2-pentenoic acid	24.55	nd	nd	1.45	nd	0.48
hexanoic acid	25.05	34.08	21.6	34.17	36.93	12.20
octanoic acid	27.87	39.69	28.88	40.17	43.75	33.41
**Ketones (7)**						
acetone	1.85	0.88	nd	0.02	nd	0.13
2-butanone	2.44	nd	0.01	0.05	0.05	0.12
2-pentanone	3.45	0.02	0.04	0.12	0.03	1.15
2-heptanone	11.56	3.64	6.59	6.71	3.59	39.82
2-nonanone	17.74	0.07	0.08	0.08	0.03	0.38
2-octene-4-one	19.49	0.03	0.03	0.01	nd	nd
5-butyldihydro-2(3H)-furanone	25.98	0.01	0.01	0.06	nd	nd
**Aldehydes (5)**						
acetaldehyde	1.50	nd	0.11	nd	nd	nd
hexanal	6.20	nd	0.20	nd	nd	nd
Octanal	15.33	0.01	0.03	nd	nd	0.05
benzaldehyde	20.10	nd	nd	0.02	0.01	0.03
2,4-dimethylbenzaldehyde	24.57	nd	0.16	nd	0.07	nd
**Alkanes (4)**						
cyclotriazane	2.16	0.03	nd	nd	nd	nd
6-methyltridecane	22.25	0.04	nd	nd	nd	nd
8-hexyl-pentadecane	24.84	nd	0.14	ns	nd	nd
2,6,11-trimethyl-dodecane	24.91	nd	nd	nd	0.26	nd
**Amides (2)**						
erucamide	32.04	0.30	nd	nd	0.68	nd
oleic acid amide	32.3	0.29	nd	nd	0.15	nd
**Phenols (2)**						
2-methoxy-3-(2-ene)-phenol	29.12	0.11	0.05	nd	nd	0.12
eugenol	29.13	nd	nd	0.17	0.63	nd
**total**		99.95	99.95	99.97	99.97	99.98

The value expressed as a percentage of total volatile compounds. nd means not detected.

**Table 2 molecules-26-02604-t002:** Performance of 10 sensors for PEN 3 electronic nose.

Sensor Number in Array	Sensor Name	General Description
S1	W1C	Aromatic compounds
S2	W5S	Reacts to nitrogen oxide
S3	W3C	Ammonia, aromatic compounds
S4	W6S	Mainly hydride
S5	W5C	Alkanes, aromatic compounds, less polar compounds
S6	W1S	Methane
S7	W1W	Sulfur compounds, otherwise sensitive to many terpenes and organic sulfur compounds, which are important for smell, limonene, pyrazine
S8	W2S	Alcohol, partially aromatic compounds
S9	W2W	Aromatic compounds, organic sulfur compounds
S10	W3S	Reacts to high concentrations, selective methane

## Data Availability

The data presented in this study are available on request from the corresponding authors.

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
