# Peer review of "Changes in Phenols, Polysaccharides and Volatile Profiles of Noni (Morinda citrifolia L.) Juice during Fermentation"

_molecules, 2021, doi:10.3390/molecules26092604_

Round 1

Reviewer 1 Report

The manuscript “Changes of Phenols, Polysaccharides and Volatile profiles of Noni (Morinda citrifolia L.) Juice during Fermentation” contains new and significant information on the effect of fermentation on phenols, polysaccharides and volatile compounds content in Noni juice. The paper's argument is built on an appropriate base of theory, concepts and other ideas. The results are presented clearly and analysed appropriately. The conclusions adequately tie together the other elements of the paper. The paper identifies clearly implications for research, practice and society. However, it should improve on keeping up with the conclusion based on strong evidence provided in their results and restrain from making assumptions that were not confirmed nor examined.

Let us clarify this.

Taste, as a potential word in the food industry, has everything to do with how our tongue and mouth interact with food and drink and the reaction that we feel is ‘taste’. Every human tongue has two kinds of receptors-one is for taste, popularly known as ‘taste buds’ which can be found all over the tongue. While the other receptor is called ‘mouthfeel’ that refers to the free nerve endings sensed all over the inside of the mouth and tongue. This receptor sends the message regarding the texture of the food or drinks we consume. In simple language, taste buds focus on whether the food is sweet, salty, bitter and sour. While the mouthfeel focuses on viscosity (i.e. body), tannins, and the overall texture of the food.

The word ‘Aroma’, often used to describe a food place or food, in particular, has two major body elements involved and that are nose and brain. It is a pleasant word used to describe odours (i.e. what we use our nose to detect). Odours are sensed in our brain by the limbic system which is an evolutionarily ancient part of our brain that also deals with emotion, behaviour, motivation and long-term memory.

Flavour as a term is very holistic in terms of food and beverage. It summarises the overall impression of a food or drink and this impression includes-aromatics, taste, and mouthfeel too. In technical language, flavour is how the brains synthesize aromas, taste, and texture as an overall experience.

In a nutshell, Flavor is the combined sensation perceived via the chemical senses (taste, smell, chemical irritation) from a food in the mouth. Taste is the sensation perceived in the mouth, more specifically on the tongue. Aroma (or smell or odor) is the sensation perceived when volatile compounds are sniffed through the nose.

When we get back to methods of this research, we can easily perceive that no sensory analysis was conducted that can give us an idea of the mouthfeel of noni juice examined, no taste buds involved, no nerve endings. Authors also say that the aim of this study was to explore the change of volatile compounds, phenols and polysaccharides and are concluding that “These 25 compounds contributed to the common flavor of fresh and fermented noni juice.” and “Although those newly produced compounds during fermentation presented low concentrations (relative contents less than 2%), they could play a key role in the flavor of noni.” and “These sulfur compounds decreased during fermentation could contribute to the improvement of flavor of noni juice.”.

To conclude, there are a lot of “could” and “should” assumptions relating to flavour and aroma of noni juice and the fact is that taste, aroma and flavour of Noni juice were not analysed at all. Circumstantial evidence based on the profile of volatile compounds is not enough for such conclusions (assumptions).  

Therefore, we recommend for the manuscript to be published after minor revisions.

Author Response

Dear Reviewer,

Thank you for giving us the opportunity to revise and improve our manuscript (Manuscript Number: molecules-1180818). According to the comments from you and the referees, we have revised carefully our manuscript. Following is the point-by-point response to all comments. In addition, all our edits to the manuscript are tracked.

Reviewer 2 Report

In the manuscript ID: molecules-1180818 authors did comparative analysis of Phenols, Polysaccharides and Volatile profiles of Noni (Morinda citrifolia L.) Juice during Fermentation. Some parts of the manuscript need to be revised to improve the flow or to add discussion. 

Here are some specific comments that can improve the manuscript:

  • The English of the manuscript is very poor and need to be revised carefully and the manuscript also contains some trivial mistakes which need to be corrected.
  • Introduction: This section needs several revisions as some paragraphs are not connected and it is difficult to follow the flow of the information. Authors could try to identify a global problem and focus on possible solutions that can lead to the aim of the study. Lines 46- 55: please, add literature to support the statements reported.
    Line 56: The aim is not clear. Please specify better: explore changes of volatile compounds, phenols and polysaccharides during what?
  • Results and discussion:
  • Section 2.1 needs more discussion of the results presented. The description of the results is very hard to follow. Authors should divide this section into subsections or paragraphs in which each group of compounds is presented and discussed separately, followed by the major conclusion on the changes of Polyphenols, Flavonoids and Polysaccharides during Fermentation. Caption of Figure 1 has to be corrected as it shows a repetition.
  • Section 2.2 This section does not describe the possible sources of the VOCs which change during fermentation. Line 164-165: Support the sentence with literature data. line 227, should the second L63  be F63? Table 1 should report the comparison beetween the experimental and theoretical linear retention times and not the rention time/min.
  • Section 2.3 is poorly written and hard to follow. Line 244: there is a repetition. Line 255: PCA performed on what? Please, specify. Figure 5 is not properly discussed and the caption is too long. In general, section 2: Results and Discussion lacks a general discussion of the results. 
  • Materials and Methods: A section should be added to list all the chemicals used in this work including the purity, company, and region. Line 305, please give the concentration of Gallic acid used for the calibration curve. Line 311: add the spectrophotometer name, model, city and country names. Line 323, the authors did not specify if the sample was stirred. Please, give also the stirring time and rpm. Line 325: Add the length of DVB/CAR/PDMS (50/30 mm) fiber, 1 or 2 cm. Line 330: Why the authors used only one column (DB-Wax, 30 m×0.25 mm×0.25 330 μm) for the aroma compounds separation? In most cases some volatile compounds which co-elute on one column can be distinguished easily on another column with different polarity. Line 335: It is necessary to use at least other two methods to identify Volatiles (i.e. determination of the linear retention indeces and the use of commercial standars, when avaiable). The  use of only the NIST library is not sufficient for volatiles identification.
  • Conclusions: Please, include the results rather providing general information.

Author Response

(The authors gave the same response as above.)

Round 2

Reviewer 2 Report

The manuscript seems improved by the recommendations and the last status is acceptable for publication.